# Immuno-Transcriptomic Profiling of Blood and Tumor Tissue Identifies Gene Signatures Associated with Immunotherapy Response in Metastatic Bladder Cancer

**DOI:** 10.3390/cancers16020433

**Published:** 2024-01-19

**Authors:** Emma Desponds, Davide Croci, Victoria Wosika, Noushin Hadadi, Sara S. Fonseca Costa, Laura Ciarloni, Marco Ongaro, Hana Zdimerova, Marine M. Leblond, Sahar Hosseinian Ehrensberger, Pedro Romero, Grégory Verdeil

**Affiliations:** 1Department of Oncology UNIL CHUV, University of Lausanne, 1015 Lausanne, Switzerland; emma.desponds@unil.ch (E.D.); marco.ongaro@unil.ch (M.O.); hana.zdimerova@seznam.cz (H.Z.); marine.leblond@unil.ch (M.M.L.); 2Ludwig Institute for Cancer Research, University of Lausanne, 1015 Lausanne, Switzerland; 3Novigenix SA, 1066 Epalinges, Switzerland; davide.croci@novigenix.com (D.C.); noushin.hadadi@novigenix.com (N.H.); sara.fonsecacosta@novigenix.com (S.S.F.C.); laura.ciarloni@novigenix.com (L.C.); sahar.hosseinian@novigenix.com (S.H.E.); pedro.romero@novigenix.com (P.R.)

**Keywords:** bladder cancer, blood biomarkers, immunotherapy

## Abstract

**Simple Summary:**

Muscle-invasive bladder cancer (MIBC) accounts for 25% of bladder cancer cases. Despite the broader usage of immune checkpoint blockades targeting the PD-1/PD-L1 axis, the response rate and survival of patients remain low for this disease. Redirecting these patients swiftly toward alternative therapeutic strategies upon immunotherapy failure should be a priority. So far, no marker allows the early outcome of the treatment to be precisely determined. The aim of our study was to determine to what extent a whole blood transcriptomic analysis could reflect the efficiency of immunotherapy in a well-established MIBC genetic mouse model. We report that it is a valuable approach to predict the response to immunotherapy in our model.

**Abstract:**

Blood-based biomarkers represent ideal candidates for the development of non-invasive immuno-oncology-based assays. However, to date, no blood biomarker has been validated to predict clinical responses to immunotherapy. In this study, we used next-generation sequencing (RNAseq) on bulk RNA extracted from whole blood and tumor samples in a pre-clinical MIBC mouse model. We aimed to identify biomarkers associated with immunotherapy response and assess the potential application of simple non-invasive blood biomarkers as a therapeutic decision-making assay compared to tissue-based biomarkers. We established that circulating immune cells and the tumor microenvironment (TME) display highly organ-specific transcriptional responses to ICIs. Interestingly, in both, a common lymphocytic activation signature can be identified associated with the efficient response to immunotherapy, including a blood-specific CD8+ T cell activation/proliferation signature which predicts the immunotherapy response.

## 1. Introduction

An immune checkpoint blockade has become a standard of treatment for several types of cancer and led to some tremendous progress in anti-tumor therapy [1]. Blocking the programmed cell death protein 1/programmed cell death ligand 1 (PD-1/PD-L1) axis has improved the prognosis of numerous patients with advanced and metastatic cancer since its approval by the FDA in 2014 [2]. However, a large proportion of treated patients fail to display clinical benefits from the blocking antibodies, and the response rate largely varies between different cancer types [3,4]. Biomarkers to predict the response to anti-PD-1/PD-L1 therapy are thus highly desirable to better identify patients who will show clinical benefits and those who should be referred to other therapeutic paths. Today, tissue-based PD-L1 expression is the only FDA-approved biomarker to guide clinicians’ treatment decision-making. Although it is generally considered that a high PD-L1 expression is associated with a favorable prognosis, its limited performance and requirement for high-quality sampling make it an imperfect biomarker of response. Furthermore, in bladder cancer, PD-L1’s association with the response to an immune checkpoint blockade remains controversial [5,6]. Alternative biomarkers that are more specific and less invasive are desperately needed.

Bladder cancer is the fourth most common cancer type in men and eleventh in women, with more than 200,000 people dying from the disease yearly (World Health Organization, WHO). A better outcome is reached at early diagnosis; however, 25% of patients are diagnosed at later stages, when the cancer has metastasized [7]. Patients with metastatic bladder cancer are directed to cisplatin-based chemotherapy as a first-line treatment, if eligible, and PD-1/PD-L1 immunotherapy as a second-line treatment [5]. However, only 15–24% of patients show durable benefit from ICIs and need to be redirected to taxane-based chemotherapy, combination ICI therapy, or newer therapeutic agents. Biomarkers that can predict the clinical outcome of anti-PD-(L)1 are urgently needed, especially for metastatic bladder cancer patient management.

Blood-based biomarkers represent ideal candidates for the development of non-invasive immuno-oncology-based assays [8]. Unfortunately, to date, no blood biomarker has been validated to predict clinical responses to immunotherapy. One major limitation to the development of blood-based assays has been the trade-off between high-depth analysis to retrieve tumor-derived signals and their cost-effective implementation in standard clinical settings. This typically limits the use of circulating tumor cell (CTC)-based analyses. One workaround is to measure the immune system’s reaction to tumor evolution rather than tumor signals themselves. We previously demonstrated the use of the circulating immune system’s transcriptional response to develop colorectal cancer screening biomarkers and identify early responses to anti-PD-1 in patients with metastatic bladder cancer [9,10,11].

In this study, we aimed to identify biomarkers associated with responses to immunotherapy in blood and neoplastic tissue to assess the potential application of simple, non-invasive blood biomarkers as a therapeutic decision-making assay compared to tissue-based biomarkers. To answer this question, we performed a bulk RNA-seq analysis of the whole-blood and matched primary tumor immune transcriptome in a genetic murine model of MIBC. Our results illustrated that although the circulating immune cells and the TME display highly organ-specific transcriptional responses to ICIs, a common CD8+ T cell activation signature can be identified in both, associated with responses to immunotherapy.

## 2. Materials and Methods

### 2.1. Mouse Model of Bladder Cancer

Tp53^Fl/Fl^Pten^Fl/Fl^ mice were obtained as previously described [12]. To initiate bladder tumor development, a volume of 5 μL of DMEM/hexadimethrine bromide (8 mg/mL) mixed with 2.5 × 10^8^ plaque-forming units of Cre-expressing adenoviral vector (#AVL(VB181004-1095pzc)-K1, VectorBuilder, Chicago, IL, USA) was surgically injected into the bladder lumen of Tp53^Fl/Fl^Pten^Fl/Fl^ mice as previously described [13].

### 2.2. Therapeutic Schedule and Treatment Groups

Therapeutic treatments were initiated 9 to 11 weeks after vector injection. In short-term trials, mice were euthanized 9 days after the beginning of the treatments, either to analyze the immune microenvironment or for an RNA sequencing analysis. The administration of anti-PD-1 (300 μg/dose, RMP1-14 clone, BioXcell, Lebanon, NH, USA) or an isotype control (IsoCT; 300 μg/dose, 2A3 clone, BioXcell) was carried out via intraperitoneal (i.p.) injections at intervals of 2 to 3 days for 8 days. One i.p. injection of anti-CD40 (100 μg/dose, FGK45 clone, BioXcell) or IsoCT (100 μg/dose, 2A3 clone, BioXcell) was performed at the beginning of the treatment. For the long-term survival experiment, 14 mice were treated with an isotype control, 9 mice were treated with anti-PD-1 monotherapy, and 30 mice were treated with the combination therapy anti-PD-1 + anti-CD40, pooled from several experiments. For the 9-day treatment intervention, 4 mice per group were treated with either anti-PD-1 or anti-PD-1 + anti-CD40. We collected blood samples at baseline and at day 9 for each mouse and collected tumor tissue from 3 out of 4 mice per group. The RNAseq data from these samples were then used for biomarker identification and response prediction modeling. In the flow cytometry validation experiments, we used between 6 and 7 anti-PD-1-treated animals and 5 anti-PD-1 + anti-CD40-treated animals, respectively.

### 2.3. RNA Sequencing and Bioinformatic Analysis

Whole bladders were harvested and processed using gentleMACS M tubes (#130-094-392, Miltenyi Biotec, Bergisch Gladbach, Germany). Total RNA was then extracted using the RNeasy Plus Mini Kit (#74134, Qiagen, Hilden, Germany) according to the manufacturer’s instructions. Mouse blood was collected from the submandibular vein 9 days after treatment and put into RNAprotect Animal Blood Tubes (76544, Qiagen), and total RNA was extracted using the RNeasy Protect Animal Blood Kit (73224, Qiagen) according to the manufacturer’s instructions. Libraries were prepared using the NEBNext Ultra II Directional RNA Library Prep Kit for Illumina (NEB). For samples extracted from blood, a globin removal step was included using the QiaSeq FastSelect globin kit (Qiagen). First-strand cDNA synthesis was followed by second-strand synthesis and the purification of double-stranded cDNA with AMPure XP beads (Beckman Coulter, Brea, CA, USA). Subsequently, cDNA was end-prepped, adaptor-ligated, and amplified through index PCR with index primers from NEBNext Multiplex Oligos for Illumina (Unique Dual Index Primer Pairs) (NEB, Ipswich, MA, USA). The samples were subjected to a total of 13 PCR cycles. Prior to PCR amplification, a qPCR was performed to determine the optimal PCR cycle number. The PCR products were purified with AMPure XP beads. To avoid index hopping on the NovaSeq sequencing platform, adapter dimers present in the libraries were excluded through size selection on 2% E-Gel EX agarose gels. The concentration of the final libraries was measured on a Fluostar Optima plate reader (BMG Labtech, Champigny-sur-Marne, France) using the Quant-iT Picogreen dsDNA assay kit (ThermoFisher Scientific, Waltham, MA, USA) at 480/520 nm, while the quality was determined on an Agilent Tapestation 4200 with the High-Sensitivity D5000 ScreenTape Assay (Agilent, Santa Clara, CA, USA). Based on these measurements, the libraries were multiplexed into a total of 9 pools. High-coverage sequencing was performed on the Novaseq 6000 system with the S4 flowcell and PE150 configuration. An average coverage of 67 million reads per sample was achieved.

### 2.4. Data Processing and Quality Check

The sequence data quality was evaluated using FastQC (version 0.11.8) combined with MultiQC (version 1.4). Atropos (version 1.1.7) was used to remove any remaining Illumina adapters or sequences longer than expected. Reads were aligned to the MOUSE genome assembly (UCSC Mus musculus mm10) along with its corresponding annotation. The program Hisat2 (version 2.2.0) was used to align a proportion of the reads to the genome using default parameters. The quantification of the transcript abundance of the RNA-Seq reads was carried out using Salmon (version 1.4.0) with default parameters. After alignment and quantification, a comprehensive QC analysis was performed, considering the following parameters: sample alignment statistic metrics such as the total number of mapped reads, secondary alignments, non-unique alignments, reads aligned to genes, alignments without any feature (intronic and intergenic), ambiguous alignments, unmapped reads, and coverage profiles. The global distribution of counts in each sample: this metric helped evaluate the distribution of gene expression levels across the samples. The proportion of features with low counts in the sample: the number of features with low counts was assessed to understand the data quality. Per-gene biotype count distribution, including residual ribosomal RNA quantification: this analysis provided insights into the distribution of counts based on gene biotypes and included the quantification of ribosomal RNA. This extensive QC process ensured that the data used for the downstream analysis were of high quality and accurately represented the biological samples under investigation.

### 2.5. Data Transformation and Exploratory Analysis

Normalization for gene length was conducted as a step downstream in our analysis of Transcripts Per Million (TPM) values. Then, we imported the gene pseudo-counts from Salmon into the R statistical computing environment (version 4.0.0), and subsequently, we applied filtering criteria, excluding genes with less than 1 count per million (CPM) across all samples and with a coefficient of variance (cv) of 100. This gene filtering process was implemented by using the filtered.data function within the NOISeq R package (version 2.31.0). Following the initial gene data treatment, we proceeded with a forward normalization step. This involved employing the variance-stabilizing transformation using the vst function, which is a feature of the DeSeq2 R package (version 1.28.1). Our primary focus for the exploratory data analysis was centered on the vst-transformed values and the selected subset of genes from NOISeq. For a comprehensive understanding of our data, we utilized a Principal Component Analysis (PCA) and scatter plots to visualize the similarities and differences among our samples. Moreover, to gain insight into the sources of variation within our dataset, we performed a Variance Partitioning Analysis (VPA) with the PVCA R package (version 1.28.0). This analysis allowed us to quantify the contributions of various confounding variables, highlighting the percentage of variation attributable to each factor.

### 2.6. Differential Expression Analysis (DEA)

We performed a comprehensive analysis of differential gene expression using three distinct methods available in the Bioconductor R packages DESeq2 (version 1.28.1), edgeR (version 3.30.3), and limma (version 3.44.3). All the methods took a read count matrix and a condition label vector as input. The parameters were set based on the guides of the corresponding R packages. A differential expression analysis was executed, accounting for factors such as batch biases and sample-specific covariates, to identify genes that exhibited significant expression changes across different conditions (details in Section 3). The results from these three methods were compared and integrated to ensure robust and reliable findings in our investigation of gene expression differences. A *p*-value cutoff ≤ 0.01 was used to identify the differentially expressed genes (DEGs) for each of the three methods, and the combination of all DEGs was used for the subsequent analysis to broaden our initial biomarker pool. Functional and network analyses of the DEGs were carried out with STRING [12] and Cluster-Profiler 4.2.2 [13] to perform an over-representation analysis (ORA), which allowed us to identify central biological pathways and biomarkers of responses. Significantly enriched ORA pathways were defined by an adjusted *p*-value ≤ 0.05. Additionally, we extrapolated the DEGs attributed to any enriched terms from the ORA results output, which allowed us to identify the functionally relevant genes among all DEGs. Plots were created with RStudio version 4.2.1, ggplot2 package version 3.4.3, and UpSetR package 1.4.0.

### 2.7. Modeling

For response group prediction modelling, three distinct groups were used, comprising 8 blood samples undergoing treatment (4 samples with anti-PD-1 and 4 samples with anti-PD-1 + anti-CD40) and 8 tumor samples (4 samples with anti-PD-1 and 4 samples with anti-PD-1 + anti-CD40). To ensure accuracy, the modeling phase employed counts after post-filtering and normalization using a variance-stabilized transformation (vst), resulting in a count matrix encompassing 15,444 genes. Acknowledging the limitation of working with a small dataset of only 8 samples, our modeling approach employed the machine learning random forest algorithm. This algorithm is renowned for its effectiveness in handling limited datasets. To evaluate our model, we implemented a 3-fold cross-validation methodology. This involved dividing the available data into two sets: 4 samples for training and 4 samples for testing. We assessed the model’s performance by plotting the True Positive Rate (TPR) against the False Positive Rate (FPR) at a sensitivity specificity threshold of 90%. To classify samples as responders or non-responders, we used a 55% probability cut-off. This systematic approach allowed us to comprehensively evaluate the model’s performance despite the constraints imposed by the small sample size.

### 2.8. Single Cell Preparation, FACS Staining and Analysis

Tumor-bearing bladders were digested for 30 min at 37 °C in complete RPMI (RPMIc, 10% FCS, and 1% penicillin/streptomycin), 0.1 mg/mL of DNase I (#D4527, Sigma, Waltham, MA, USA), and 1 mg/mL of Colagenase I (#17100017, ThermoFisher Scientific, Waltham, MA, USA). Tissues were then mashed through a 70 μm cell strainer. Leucocytes were isolated through sample centrifugation in a density gradient of 40%/70% Percoll for 30 min at 2000 rpm. Cells from the interphase were harvested and washed in RPMIc before staining.

Blood was collected from the submandibular vein, and erythrocytes were eliminated with homemade red blood lysis buffer (H_2_O, 156 mmol/L of NH_4_Cl, 12 mmol/L of NaHCO_3_, and 0.1 mmol/L of EDTA) for 2 min before staining. FcγR blocking was then performed at room temperature (RT) with anti-CD16/32 (1/1000, #101320, Biolegend, San Diego, CA, USA). Following extracellular marker staining, cell viability was assessed using the Zombie NIR Fixable Viability Kit (#423105, Biolegend) following the manufacturer’s instructions. Cells were then fixed and permeabilized with the Foxp3 Transcription Factor Staining Buffer Set (#00-5523-00, eBiosciences, Villebon-sur-Yvette, France) according to the manufacturer’s instructions. Intracellular staining was carried out in permeabilizing buffer from the Foxp3 Transcription Factor Staining Buffer Set. The specific antibodies used for the flow cytometry analysis include SparkUV387 CD8 at 1:200 (53.6.7 from Biolgened), BUV496 CD4 at 1:200 (GK1.5 from BD, Franklin Lakes, NJ, USA), BUV737 CD44 at 1:100 (IM7 from Invitrogen, Waltham, MA, USA), BUV805 CD45.2 at 1:50 (104 from BD), eFl506 CD3 at 1:50 (17A2 from Invitrogen), BV570 CD45R at 1:50 (RA3-6B2 from Biolegend), BV650 CX3CR1 at 1:200 (SA011F11 from Biolegend), FITC CD48 at 1:100 (HM48-1 from Biolegend), PE/Dazzle594 CD223 at 1:200 (C9B7W from Biolegend), PECy7 CD279 at 1:100 (29F.1A12 from Biolegend), APC CD161 at 1:100 (PK136 from Invitrogen), eFluor450 FoxP3 at 1:100 (FJK-16s from Invitrogen), and BV421 Ki67 at 1:100 (16A8 from Biolegend). Data were acquired on a Cytek AURORA machine and analyzed with FlowJo software V10.

## 3. Results

### 3.1. Identification of Biomarkers Correlating with Immunotherapy Response in Bladder Cancer

To identify blood biomarkers correlating with the clinical benefits of immune checkpoint inhibitor (ICI) therapy, we used a genetic mouse model recapitulating the various steps of bladder cancer (BC) [12,13]. Briefly, Tp53^Fl/Fl^Pten^Fl/Fl^ mice received adenoviral vectors encoding for cre-recombinase in the bladder, leading to the development of tumors that progressed from non-muscle-invasive (NMIBC) to muscle-invasive BC (MIBC) at the metastatic stage 10 to 13 weeks after tumor induction [12,13]. The transcriptomic profiles obtained from these mice show strong similarities to a basal-squamous subtype, representing 35% of MIBC patients [12]. This model is resistant to anti-PD-1 blocking monotherapy, but dual anti-PD-1/anti-CD40 therapy restores the anti-PD1 blockade potential and strongly increases mouse survival [12] (Figure 1A). To identify biomarkers correlated with responses to immunotherapy, we induced tumors in Tp53^Fl/Fl^Pten^Fl/Fl^ mice and waited for them to progress to the MIBC/metastatic stage. We then split the cohort into two groups: one group receiving monotherapy (anti-PD-1, n = 4) and one group receiving combination therapy (anti-PD1/anti-CD40, n = 4) (Methods). Both groups were treated with corresponding therapy regimens until sacrifice. Whole-blood samples were collected after the treatment from the submandibular vein, and bladder tissue was also collected at the end of the treatment, showing increased weight when tumors were induced but no difference in weight between the two treatment conditions 9 days post-treatment (Figure 1B). A subset of the biological samples were analyzed by RNA-sequencing (RNA-seq), and differential expression analyses (DEAs) were performed between the different treatment groups to identify differentially expressed genes (DEGs) in the blood and tumors, respectively, at the on-treatment timepoint (day 9). We performed DEAs between the on-treatment (day 9) blood samples from the responder mice (treated with combination therapy) and the non-responder mice (treated with monotherapy) (Figure 2A). The DEA results led to the identification of 507 genes that were found to be differentially expressed between the two treatment groups (see Section 2 for the DEA details). Among these, 257 genes were found to be upregulated in the blood from the mice that received the combination therapy, while 250 genes were found to be downregulated (Figure 2A, Appendix A). In the responder mice’s upregulated genes, a bioinformatic analysis using the gene ontology annotations of the over-representation analysis (GO ORA) showed a strong enrichment in terms associated with cell proliferation (the regulation of the cell cycle process, the regulation of mitotic sister chromatid separation, mitotic cell cycle phase transition, and others; Figure 2B). Among the downregulated genes, we found an enrichment in GO terms associated with RNA metabolism (RNA splicing, mRNA processing; Figure 2C). To obtain a more complete picture of the pathways emerging from our data, we used a Protein–Protein Interaction Network Functional Enrichment Analysis (STRING) to put forward other potential networks. For the upregulated genes, on top of genes involved in proliferation (Aurkb, Top2a, Cdca8, Ccnb; Figure 2D), we found genes associated with lymphocyte activation (Pdcd1, Lag3, Cd48, Cx3cr1, Ccl5, or Gzmk; Figure 2E). We also found a small cluster of genes associated with type I IFN system activation (Ifi44, Oasl2, Isg15; Figure 2F). Among the downregulated genes, this analysis also led to the identification of a cluster of genes involved in chaperone-mediated protein folding (Hsph1, Hspa8, Dnajb1, Fkbp4, Fkbp5, Ptges3) or the regulation of cell death (Hsp90b1, Herpud1, Dnaja1, Zbtb16; Figure 2G). Altogether, this analysis of whole-blood RNA revealed the presence of proliferating immune cells expressing lymphocyte activation markers associated with effective immunotherapy in the MIBC setting.

### 3.2. Tumor Analysis Outlined the Complexity of the TME Response to Immunotherapy

To identify tumor tissue biomarkers associated with the response to immunotherapy, we performed DEAs between the responder mice treated with combination therapy and the non-responder mice treated with monotherapy (Figure 3A, Appendix A). Using the same DEA approach as for the blood sample analysis, 269 genes were found to be differentially expressed. Among the genes highly expressed after the combination treatment, the GO ORA bioinformatic analysis showed enriched terms for T cell and myeloid cell activation as well as the production of cytokines (Figure 3B). This enrichment was confirmed with the STRING analysis, with an enrichment for genes associated with T cell activation (Tnfrsf9, Lilrb4, Sirpb1c, SIrb1d), inflammatory response (Ccl6, Ccl9, Ccr1, Ccr2, Ccr5), and myeloid cell activation (Pirb, Lilrb4, Fcgr3, Fcgr4) (Figure 3D). Additionally, the STRING analysis highlighted the upregulation of genes associated with oxidative phosphorylation (Cox5a, Cox7a1, Cox7b, Cycs) and mitochondrion organization (Bnip3, Mief2, Ndufaf5, Ndufb8, Dna2) (Figure 3E). Among the downregulated genes, the enriched GO terms comprised smooth muscle contraction, the vascular process in the circulatory system, blood vessel diameter maintenance, and wound healing (Figure 3C). Overall, this analysis showed the more complex nature of the response found in the tumor, with increased lymphocyte activation, and its effect on myeloid cells and other cell types present in the tumor microenvironment.

### 3.3. Shared Transcriptional Response to ICIs between Blood and TME

To assess the potential and extent of overlap between the tissue and peripheral blood transcriptional responses to immunotherapy, we extracted the overlap between the differentially expressed gene lists from the responder and non-responder comparisons for both the blood and tumors (Figure 3F). By comparing the previously identified DEG lists (507 DEGs in blood and 269 DEGs in tumors), we only found a few genes to be commonly regulated in the blood and the bladder (Tinagl1, Adgre4, Mphosph9, Nacad, Gm7334, Gm4841, Galnt3, Akap12, Gm4117, C920009, B18Rik) (Figure 3F). Despite the low number of shared genes, we hypothesized that similar biological processes might be affected in both blood and tumor tissue. The treatments target immune cells, and we reasoned that immune cells will be affected similarly, to some extent, in both organs, even though this signal will be harder to find in a bladder tumor, as many non-immune cell types, including tumor cells, are also present. Therefore, we broadened our research to the enriched GO terms of both the blood and tumors. This approach highlighted shared biological pathways, and specifically, we found 11 common GO terms in the upregulated genes (Figure 3G) and none in the downregulated genes (Figure 3H). Interestingly, the commonly enriched GO terms from the blood and bladder included several terms associated with T cell biology (the regulation of cytokine production, cell adhesion, adaptive immune response, T cell activation, the regulation of leukocyte cells,), proliferation (the regulation of nuclear division), and inflammation (inflammatory response), indicating that the central processes of the therapeutic response in the responder mice can be detected in both the liquid and tissue biopsies.

### 3.4. Prediction of Immunotherapy Response Using Blood or Bladder Signature 

We determined clear biological signatures emerging from the bulk RNA-seq from the blood and tissue, which varied between the responder and non-responder animals (Figure 2 and Figure 3). To determine whether these signatures could be predictive of the response, we built machine-learning-based therapy prediction models from either the blood- or tumor-derived signatures. Receiver operating characteristic (ROC) curves were generated, and the model performance was assessed by evaluating the area under the curve (AUC). First, we examined whether two broad gene signatures derived from the overall GO ORA analysis could be predictive of the response (Figure 4A,B). The broad GO signatures were prepared by extrapolating the DEGs that were attributed to any enriched terms, which allowed us to identify which genes were functionally relevant to the enriched pathways. Both signatures showed the ability to separate the responder mice from the non-responder mice when tested on blood or tumor samples. Then, we assessed the biology-specific signatures that were previously identified (Figure 2 and Figure 3). Specifically, we created three predictive models based on blood-derived signatures linked to cell proliferation, T cell activation, and type I interferon (Figure 4A) and two models based on the tumor-derived signatures’ oxidative phosphorylation and tumor inflammation (Figure 4B). The blood-derived signature models showed a trend toward better performance than the tumor-derived ones, highlighting the potential of capturing gene expression signals in whole-blood samples related to responses to ICI therapy. Last, we tested the performance of literature-derived signatures associated with responses to immunotherapy in this specific pre-clinical intervention setting. We queried a pan-cancer-derived signature and created a predictive model based on its gene list (Figure 4C) [14]. Interestingly, this model allowed us to discriminate between therapy responses with a similar ability to the blood-derived models.

### 3.5. CD8 T Cells Are the Main Cell Type Showing Increased Proliferation and Expression of Lymphocyte Activation Markers

The cell cycle signature and lymphocyte activation were among the most efficient signatures to predict immunotherapy responses in our settings. To validate the bioinformatic analysis and machine-learning-based predictive models and understand which cell types are responsible for these signatures, we monitored the immune responses of several immune cell types in the blood and the bladder by flow cytometry. Focusing on lymphocytes, we did not measure significant changes in the proportion of B cells (B220+), CD4 T cells (CD3+ CD4+), regulatory T cells (Foxp3+ CD3+ CD4+), or NK cells (CD3- NK1.1+) but found an increased number of CD8+ T cells in the blood and bladder after combined treatment (Figure 5A). We then focused on the expressions of some lymphocyte activation molecules found in the lymphocyte activation cluster (Figure 2). In the blood, we found significant changes in the expressions of PD-1, CX3CR1, LAG3, and CD48 in several of the studied populations, but only CD8+ T cells consistently upregulated the four molecules, with the highest fold change (Figure 5B). To assess cell proliferation, we stained the blood samples with anti-Ki67. CD4+ T cells and CD8+ T cells had increased staining for Ki67, but CD8+ T cells showed the highest fold change again (2.5-fold vs. 1.4-fold). Additionally, we found higher levels of Ccl5 in the serum of the responsive mice compared to the unresponsive mice (Figure 5C), as found in our transcriptomic analysis (Figure 2F). Altogether, the number of CD8+ T cells increased after successful immunotherapy, together with an increased expression of activation markers and increased proliferation, and they are therefore likely to be the main cell population contributing to the gene expression signal measured in the blood.

## 4. Discussion

Monitoring the response to immunotherapy quickly, reliably, and efficiently remains an unmet need to redirect patients rapidly towards other treatments in cases of response failure. In patients, it is much more convenient to obtain blood samples than biopsies of the tumors, making blood the compartment of choice to study. Moreover, the circulating immune compartment, together with extracellular vesicles and low-frequency circulating tumor cells, are exquisitely sensitive sensors integrating the systemic responses to primary tumors and metastases. This represents a unique advantage over the limited sampling offered by tumor biopsies. To understand whether we could monitor an efficient response to immunotherapy in the blood, we used a well-characterized genetic model of muscle-invasive bladder cancer that recapitulates the various steps of the human disease [12]. In this anti-PD-1-resistant model, a combination of agonistic anti-CD40 together with an anti-PD-1 immune checkpoint blockade induced a good immune response, leading to tumor control through the recruitment of newly activated CD8+ T cells in the tumor and a modification of the tumor microenvironment [12]. The comparison of the blood bulk RNAseq samples from the responding and non-responding mice led to the identification of a clear subset of genes associated with the response, comprising genes associated with proliferation, lymphocyte activation, and Type I IFN activation. All three of these signatures could predict the response in our model. These data were confirmed at the protein level for some activation markers and for proliferation. It appeared that CD8+ T cells are the major cell type responding to the treatment, confirming what we already published in this mouse model [12]. Previously, some genes associated with cell proliferation were shown to be upregulated in patients with bladder cancer responding to anti-PD-1 treatment [9]. Interestingly, LAG3, which we found to be upregulated specifically on CD8+ T cells after immunotherapy, was associated with the response to anti-PD-1 immunotherapy within the tumors of patients with urothelial carcinoma [15]. This suggests that the T cell response in the blood of patients with bladder cancer can be a sensitive readout that also predicts an efficient response to immunotherapy. Additionally, IFN-responsive genes such as Ifi44 and Isg15 (Figure 2F) are part of a cluster of genes identified in the Cancer Genome Atlas and four immunotherapy cohorts as predictors of the evolution of the disease and the response to immunotherapy [16]. The ability of whole-blood immunotranscriptomics to detect the increased expression of these genes in the blood constitutes an interesting perspective for response prediction. When looking at bulk RNAseq in the bladder, we found fewer genes differentially regulated and a more complex signature, consistent with the complexity of the tumor microenvironment composed of tumor cells, stromal cells, vascular cells, and immune cells [17]. T cell activation is still found as an overrepresented GO term, but none of the activation markers found in blood were also found in the bladder. Our work has several limitations. The first one is that we compared two different therapies as responder and non-responder treatments. Ideally, we should have compared a similar treatment with two different outcomes. As all mice respond to the anti-CD40/anti-PD-1 combo and none to anti-PD-1 alone, this was not possible in our study. The second limitation is that we worked with a mouse model with a limited number of animals for the transcriptomic study. We recognize that the limited dataset of only four samples for training and four samples for testing may not provide adequate grounds for a thorough evaluation of a model’s performance using ROC curves. Assessing a model’s performance robustly through ROC curves typically demands a more extensive dataset for both training and testing purposes. This highlights the importance of acknowledging the need for larger datasets in future evaluations to ensure a more comprehensive analysis of the model’s performance. Yet, our model is very reproducible and well characterized, which allowed us to answer our questions with this limited number of animals. Lastly, we studied the response in blood at a given time point, day 9 post-treatment. A longitudinal analysis would be interesting investigate for how long our markers of interest can be detected and associated with the response to the treatment. The next step will be to translate our observations to patients by monitoring blood samples post-immunotherapy treatment and to associate them with treatment outcomes to obtain a more precise list of biomarkers useful for predicting responses to immunotherapy in bladder cancer to rapidly redirect unresponsive patients toward alternative therapeutic strategies.

## 5. Conclusions

Overall, monitoring immune responses in whole blood is a valuable approach to determining whether a patient is responding or not to immunotherapy. A longitudinal study of patients should be performed to confirm the potential of this monitoring, but it would also open up the possibility of rapidly knowing how the patient is responding to immunotherapy and whether alternative treatments should be started. This approach is amenable to robust process development, upscaling, and integration into clinical trials and oncology clinical practice.

## Figures and Tables

**Figure 1 cancers-16-00433-f001:**
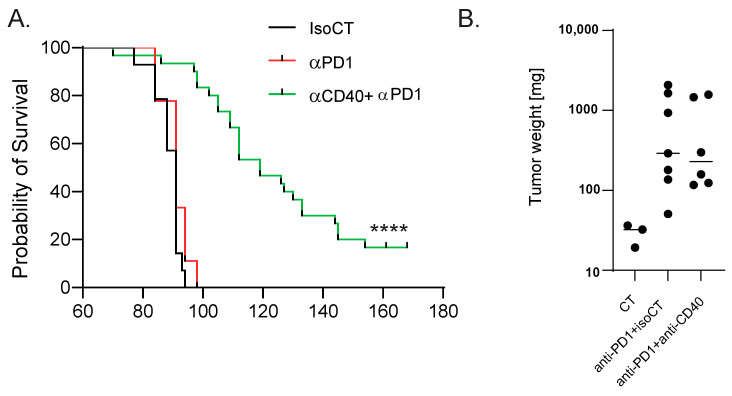
Mouse model of anti-PD-1-resistant MIBC. (**A**) Survival curves of control MIBC mice treated with isotype control Ab (IsoCT, black, n = 14) and mice treated with either anti-PD-1 (red, n = 9) or anti-PD-1 + anti-CD40 (green, n = 30). **** *p* < 0.001 by log-rank test. (**B**) Bladder weight 9 days post-treatment for the indicated conditions. Each dot represents the bladder from one mouse (2 pooled experiments).

**Figure 2 cancers-16-00433-f002:**
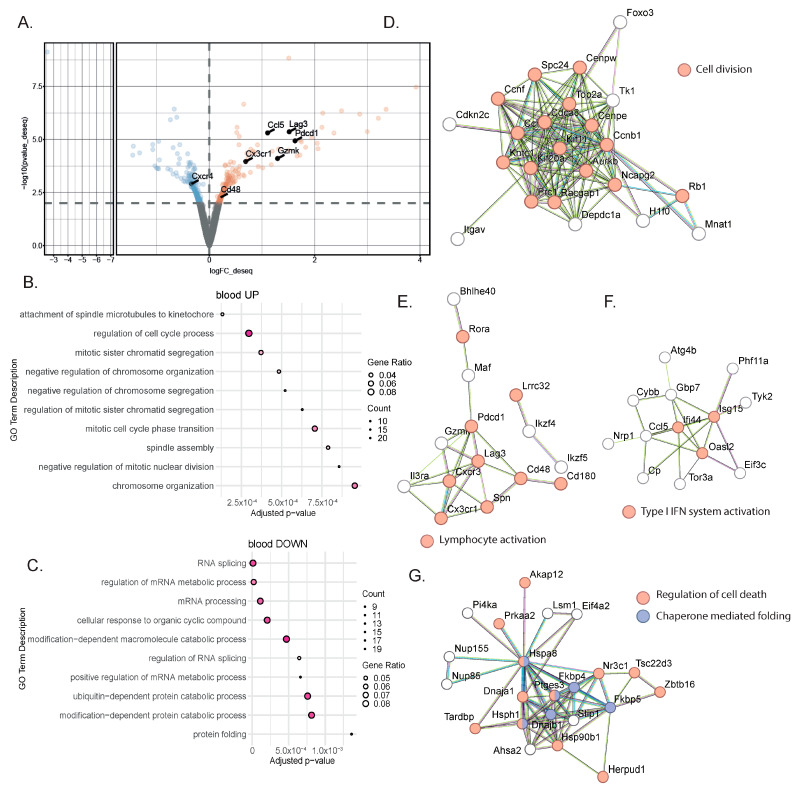
Gene signatures in the blood characterized a successful response to immunotherapy. (**A**) Representative volcano plot of the differentially expressed gene analysis using the DEseq2 method in bulk RNA-seq comparing whole blood from MIBC mice treated with anti-PD-1 + anti-CD40 (n = 4) to those treated with anti-PD-1 alone (n = 4). Statistically significant differentially expressed genes (*p*-value ≤ 0.01) between the two treatment groups at 9 days post-treatment are highlighted (downregulated genes in blue; upregulated genes in orange). (**B**,**C**) Gene ontology overrepresentation analysis (ORA) performed on upregulated (**B**) or downregulated (**C**) genes from MIBC mice treated with combination therapy compared to anti-PD-1 alone. The top 10 significant pathways are shown (based on an enrichment adjusted *p*-value ≤ 0.05) (**D**–**F**) STRING analysis on upregulated genes from MIBC mice treated with combination therapy compared to anti-PD-1 alone. Most biologically relevant clusters are shown. (**G**) STRING analysis on downregulated genes from MIBC mice treated with combination therapy compared to anti-PD-1 alone.

**Figure 3 cancers-16-00433-f003:**
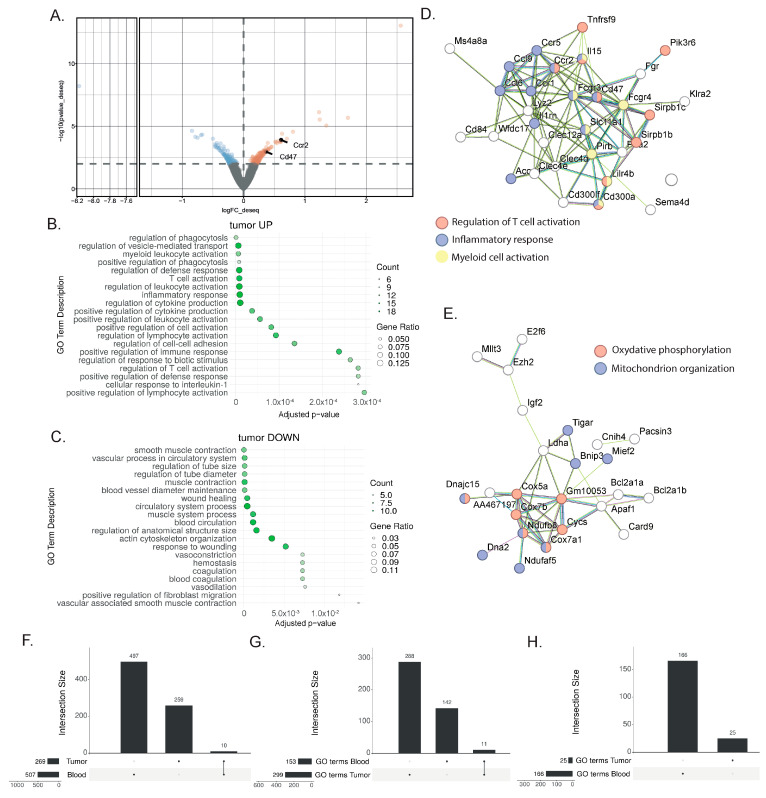
Gene signatures in the bladder after a successful response to immunotherapy. (**A**) Representative volcano plot of the differentially expressed gene analysis with the DEseq2 method in bulk RNAseq comparing whole bladder from MIBC mice treated with anti-PD-1 + anti-CD40 (n = 3) to those treated with anti-PD-1 alone (n = 3) 9 days post-treatment. Differentially regulated genes (with a *p*-value ≤ 0.01) are highlighted (downregulated genes in blue; upregulated genes in orange) (**B**,**C**) Gene ontology overrepresentation analysis (ORA) performed on upregulated (**B**) and downregulated (**C**) genes from MIBC mice treated with combination therapy compared to anti-PD-1 alone. The top 10 significant pathways are shown (based on an enrichment adjusted *p*-value ≤ 0.05) (**D**,**E**) STRING analysis of upregulated genes from MIBC mice treated with combination therapy compared to anti-PD-1 alone. Most biologically relevant clusters are shown. (**F**) Upset plot comparing the differentially expressed genes (DEGs) in tumor and blood biopsies, indicating an overlap of 10 genes. (**G**,**H**) Upset plot showing the number of significantly enriched gene ontology (GO) terms found by performing an overrepresentation analysis (ORA) of (**G**) the upregulated DEGs or the (**H**) downregulated DEGs in blood or tumors. In the upregulated pathways (**G**), 11 shared GO terms were found.

**Figure 4 cancers-16-00433-f004:**
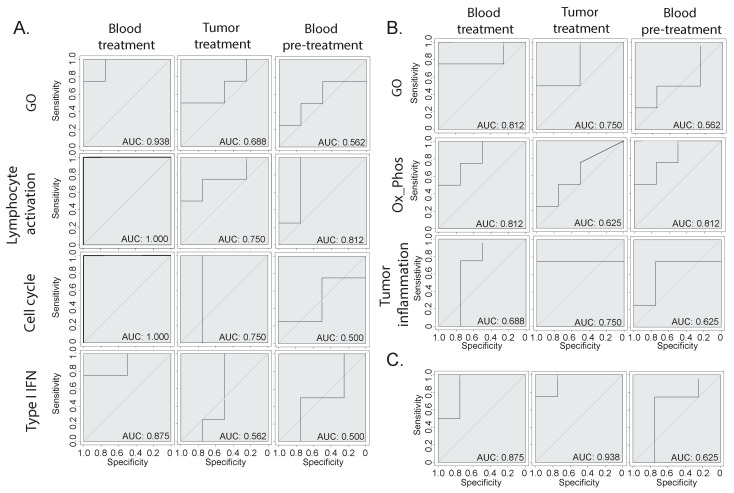
Therapy response prediction modeling. (**A**,**B**) Receiver operating characteristic (ROC) curves of therapy predictive models based on different gene signatures derived from the (**A**) blood or (**B**) tumor gene expression analysis. The GO signatures include all genes that were attributed to the enriched GO terms, while the other signatures have been manually annotated based on specific biologically relevant enriched processes. (**C**) ROC curve of a model based on a publicly available pan-cancer signature of response to immunotherapy [14].

**Figure 5 cancers-16-00433-f005:**
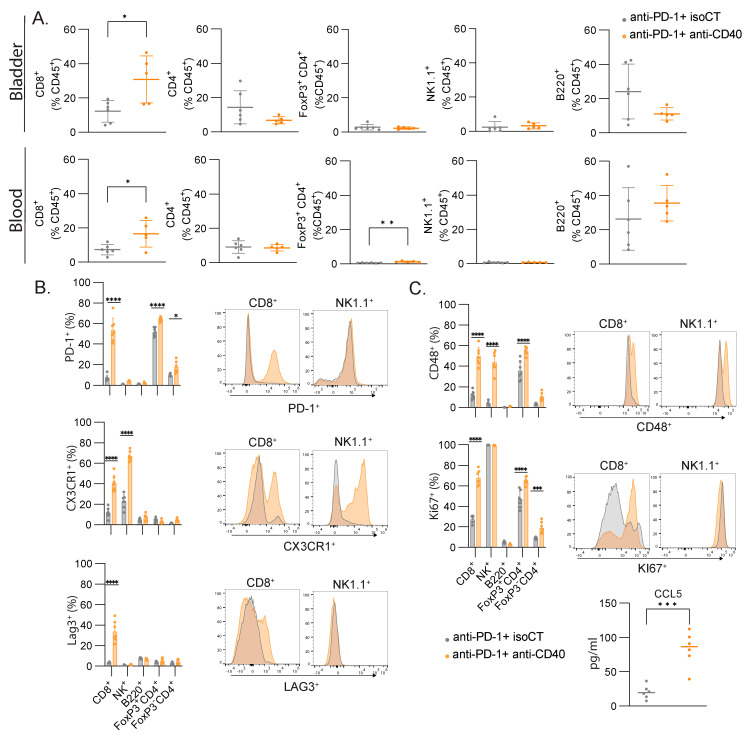
CD8+ T cells are the major contributors to the gene signatures found in blood after immunotherapy. (**A**) Relative proportion of all immune cells (CD45+) with mean percentage ± SD of the indicated population of lymphocytes in the bladder and the blood. (**B**) Representative FACS histograms and percentages of the indicated markers for the indicated lymphocyte populations. (**C**) Concentration (pg/mL) of CCL5 in the serum of mice. Mice were treated at MIBC stage with either anti-PD-1 and isoCT isotype control (grey, n = 7) or anti-PD-1 and anti-CD40 (orange, n = 7). Analyses were performed 9 days after the beginning of the treatments. For scatter plots, each dot represents an individual mouse, and bars represent the mean. * *p* < 0.05; ** *p* < 0.01; *** *p* < 0.005, **** *p* < 0.0001 by ANOVA test, followed by a Sidak test.

## Data Availability

Data can be accessed through the following DOI:10.5281/zenodo.10245156.

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
