# Peer review of "Immuno-Transcriptomic Profiling of Blood and Tumor Tissue Identifies Gene Signatures Associated with Immunotherapy Response in Metastatic Bladder Cancer"

_cancers, 2024, doi:10.3390/cancers16020433_

Round 1
Reviewer 1 Report
Comments and Suggestions for Authors
In the present work Emma et al., aimed at identifying biomarkers associated with response to immunotherapy in blood and in neoplastic tissues, derived from murine models. The authors have performed bulk RNA-seq analysis of whole-blood immune-transcriptome and matched primary tumor in a genetic murine bladder cancer model. The present work is in the right direction as it addresses a very interesting tumor type. The present work is interesting and it has merit for publication, after addressing some major issues.
In the “Materials and Methods” section, the number of mice was nowhere to find. Also, what was the reference in the overall experimentation?
An in-text ethics statement is missing.
In general, all figures should be close to their respective text. In addition, all figures are of poor quality. Legends, axes, descriptions are barely visible (in some cases text was to be magnified by >200% to be able to see legends and figure details). Figures should be significantly improved.
In the “Results” section, Figure 1B, should be moved to the “Materials and Methods” section, along with the respective text. This section describes the methodology followed. The authors could also expand this figure and present their complete rationale in a diagrammatic form. Further on, the authors state “…we hypothesized that similar biological process might be affected both in blood and tumor tissue.” The concept is very interesting, yet why is this? Since the transcriptomic profile comes from two different cell types, why is it probable to manifest commonly regulated genes? In what sense, does the tumor expressional profile can affect immune cell expressional profile? Why is it expected for a tumor cell and lymphocytes to manifest similar (or co-regulated) expressional profiles? Please elaborate on this. Finally, the authors did not mention how they came up with the list of commonly regulated genes. What was the methodology used?
A bioinformatics analysis is missing. The authors could expand their analysis using machine learning analysis such as hierarchical clustering, or k-means to analyze and present the classification of gene expression.
I suggest to mention the limitations of the study. What can be done better, or what are the next steps?
Finally, the authors should highlight their results and mention how their findings could prove useful for the treatment of bladder cancer. How is it possible to use combinatorial immune therapy in order to treat bladder cancer?
Comments on the Quality of English LanguageUnfortunately, due to technical reasons, I don’t have access to the Turnitin tool momentarily, in order to check for plagiarism. Therefore, I wasn’t able to check for plagiarism.
Author Response
Please see the attachement

Reviewer 2 Report
Comments and Suggestions for Authors
The article is undoubtedly interesting, presented by a combination of modern clinical approaches with bioinformatic analysis and thorough statistical analysis. It should be noted that this work has significant clinical significance. However, the authors practically did not express the final idea. What is recommended in the final of this preliminary study? Where are the recommendations for additional work? The work needs to be revised in terms of the practical significance of the study, which needs to be added to the abstract and conclusion.
Comments on the Quality of English LanguageThe article is undoubtedly interesting, presented by a combination of modern clinical approaches with bioinformatic analysis and thorough statistical analysis. It should be noted that this work has significant clinical significance. However, the authors practically did not express the final idea. What is recommended in the final of this preliminary study? Where are the recommendations for additional work? The work needs to be revised in terms of the practical significance of the study, which needs to be added to the abstract and conclusion.
Reviewer 3 Report
Comments and Suggestions for Authors
A very interesting work showing the response to the therapy in an animal model. However, some issues should be clarified:
1) the authors should specify how many mice there were in which groups and how many of the sample was used for each analysis. The authors provide information in the text, in the paragraph regarding modeling that there were 6 samples, but not in the section "Materials and methods". This is not sufficient information and they should precisely provide this information for each test performed.
2) moreover, the authors practically do not write about the limitations of research conducted in such small groups. Apart from one fragment, there are no limitations to this research.:" To ensure the robust evaluation of our model, we implemented a 3-fold cross-validation methodology, partitioning the available data into two sets of 3 samples each for training and testing. This approach allowed us to comprehensively assess the model's performance within the constraints of the small sample size"
3) there is also the lack of information regarding the expression analysis, what was compared with what, the lack of results of these studies and the lack of information, for example in the supplementary materials, which genes were among the significant results and how the statistical calculations were carried out (statistics is shown only graphically)
4) it is not clear to me how ROC curves were calculated for such small groups
5) data from these analyzes should be made available in public databases and a link/address should be provided where these results can be find and verified
Round 2
Reviewer 3 Report
Comments and Suggestions for Authors
The authors made the necessary corrections.
Author Response
We thank you for your comment.